# Digital Technologies in Diagnosing Solitary Median Maxillary Central Incisor Syndrome

**DOI:** 10.3390/children13010011

**Published:** 2025-12-20

**Authors:** Katarzyna Cieślińska, Karolina Karbowska, Katarzyna Zaborowicz, Barbara Biedziak

**Affiliations:** Department of Orthodontics and Facial Malformations, Poznan University of Medical Sciences, Bukowska 70, 60-812 Poznań, Poland; karbowska.dentist@gmail.com (K.K.); kzaborowicz@ump.edu.pl (K.Z.); biedziak@ump.edu.pl (B.B.)

**Keywords:** SMMCI syndrome, development, incisor

## Abstract

Solitary Median Maxillary Central Incisor Syndrome is unique congenital developmental defect affecting midline structures of the head and the body. The prevalent symptom is a solitary median incisor of the maxilla in primary and secondary dentition, which is positioned exactly in the midline of the alveolus. Other abnormalities that are characteristic of the syndrome include holoprosencephaly, nasal cavity anomalies, cleft palate–lip, hypotelorism, and microcephaly. It is estimated to occur in 1:50,000 live births, with female gender predilection. The cause of the syndrome is related to midline defects in the migration and connection of the prechordal mesoderm between the 35th and 38th days post-conception. Early diagnosis of SMMCI is important for practicing orthodontists, as it may be a symptom of other developmental abnormalities. The aim of this study is to report a case of SMMCI syndrome in a patient treated in the Department of Orthodontics and Facial Malformation of the University of Medical Sciences in Poznan.

## 1. Introduction

Solitary Median Maxillary Central Incisor Syndrome (SMMCI) is a rare congenital disorder that affects the midline structures of the body. The abnormalities are most commonly observed in the head and facial region. This syndrome occurs more frequently in females than in males. Hall et al. [1] originally described this condition as “Solitary median maxillary central incisor syndrome, short stature, and choanal atresia/midnasal stenosis.”

However, the name is now commonly shortened to just the first part, “Solitary Median Maxillary Central Incisor Syndrome” (SMMCI), as not all cases present with the additional associated abnormalities. Using the full description of the tooth in the name is important, as it emphasizes its unique shape and position, which are the hallmark features and the most easily recognizable clinical signs of the syndrome. SMMCI was first described in 1958 by Scott [2]. Solitary Median Maxillary Central Incisor Syndrome (SMMCI) is a relatively rare disorder. There are only a limited number of scientific publications on this condition in the medical literature. According to Poelmans et al. [3], the estimated prevalence of SMMCI is 1 in 50,000 live births, and it occurs more frequently in females. Hall reported that the incidence is higher in cases of stillbirths and miscarriages. Interestingly, all documented and studied cases of cyclopia have been found to involve the presence of an SMMCI tooth [4].

In recent years, reports have been published on the genetic background as the etiology of SMMCI. The most common cause is heterozygous mutations in the SHH (sonic hedgehog) gene, located on chromosome 7q36 [5]. These abnormalities arise between the 35th and 38th days of embryonic development as a result of defective fusion of the mesoderm along the midline. A characteristic feature of the syndrome is the presence of a single maxillary central incisor located in the midline of the oral cavity. The tooth has a symmetrical shape and appears in both primary and permanent dentition. Other anatomical features commonly observed in affected individuals include: a poorly developed philtrum, absence of the upper labial frenulum, underdeveloped or absent incisive papilla, palatal hypoplasia or absence of the palatal suture anterior to the incisive foramen, and a thickened bony ridge on the hard palate [6].

Other characteristic features of the syndrome include choanal atresia, midface hypoplasia, narrowing of the pyriform aperture, hypotelorism, cleft of the primary and/or secondary palate, microcephaly, cyclopia, holoprosencephaly, and intellectual disability [7]. Early diagnosis is based on ultrasound examination between the 18th and 22nd weeks of pregnancy or on prenatal genetic testing, although the latter is rarely performed. Postnatal diagnosis should include cephalometric analysis, which often reveals abnormalities in the sella turcica, potentially indicating pituitary gland dysfunction [8].

### Aim

The aim of the study was to assess the morphology of the occlusion using digital imaging techniques in the analysis of a patient case.

## 2. Materials and Methods

The case description was based on dental and general medical history, radiological documentation, as well as extraoral photographs and intraoral scans. The material was obtained during the patient’s visits to the Orthodontic Clinic of the Department of Orthodontics and Facial Malformations in Poznań.

### Case Description

A 10-year-old patient presented to the Department of Orthodontics and Facial Abnormalities in Poznań. He was referred by a general dentist for orthodontic consultation. A general medical interview was conducted with the child’s parents, during which it was established that the pregnancy was uneventful. The delivery was induced at term and proceeded without complications. The birth weight was 3180 g, and the boy was born healthy, scoring 10 points on the Apgar scale. The child experienced difficulties with breathing and feeding; the sucking reflex was impaired. He had to be on bottle feeding from the first days of life, and the parents reported that he accepted only one specific type of pacifier. The child breathed through his mouth and snored. At four months of age, the pediatrician diagnosed increased muscle tone and recommended physiotherapy. Home exercises were introduced and followed up until muscle tone improved. The child was generally healthy and, apart from typical childhood infections during the preschool period, did not suffer from any significant illnesses. Due to disordered breathing patterns, the patient was under the care of an ENT specialist, who diagnosed narrowing of the nasal passages in the structure of the upper airways (Figure 1).

No other abnormalities were found in the structure of the throat, nose, or ears. In the interview, the parents reported that mouth breathing persists to this day, although it has been partially corrected with removable orthodontic treatment. Since starting orthodontic therapy, the child sleeps better at night and no longer snores. The child’s intellectual development has been normal and appropriate for his age. He does not suffer from any systemic diseases. A family history of dental anomalies was excluded during the interview. Primary tooth eruption began around the 6th month of life. In the deciduous dentition, a single maxillary central incisor was present. After the replacement of the primary incisor with a single permanent central incisor, the child was referred to an orthodontist by the general dentist. Dental trauma to the incisors was ruled out during the dental interview. The presence of similar dental abnormalities in patient’s relatives was also excluded. Extraoral facial symmetry and proportions are normal. No external anomalies were observed. The boy exhibits reduced tension of the orbicularis oris muscle and mixed mouth-nasal breathing. Photographic documentation of the face and intraoral scans of the dentition were obtained (Figure 2, Figure 3, Figure 4 and Figure 5).

Scans were performed using the 3Shape TRIOS Intraoral Scanner (3Shape, Copenhagen, Denmark), which enabled the creation of virtual models of the patient’s dentition (Figure 6, Figure 7 and Figure 8).

In the intraoral examination, a poorly developed upper labial frenulum and incisive papilla were observed, along with maxillary constriction and a prominent midline palatal ridge. The upper dental arch had a V-shaped form (Figure 9).

The solitary median maxillary central incisor has a symmetrical crown shape and is positioned in the midline of the maxillary arch. The dimensions of the central incisor are 9.8 × 8.1 mm. The crown does not exhibit the Muhlreiter angle (Figure 10 and Figure 11).

According to Andrews’ keys to occlusion and the rule of negative crown angulation (stating that anterior teeth crowns should be tilted mesially at an angle of 5 degrees), the single central incisor does not show appropriate mesial inclination (Figure 12 and Figure 13).

The patient’s dental occlusion scans were also registered (Figure 14, Figure 15 and Figure 16).

Radiological examination confirmed the presence of a single central incisor in the midline, with a normally developed root and a single pulp chamber (Figure 17).

The patient did not have a crossbite, but due to the asymmetry of the palate, the symmetry of the condyles and mandibular rami was measured on the ortopantomographic image. The analysis was performed following the method described by Habets et al. [9]. The height of the OPG image of the condyle corresponded to the distance H1/H2 and the height of the OPG image of the ramus corresponded to the distance H2/H3 (Figure 18). To assess the asymmetry between the right (R) and left (L) condyles and rami (asymmetry index), the formula │(R − L)/(R + L)│ × 100 was used. An asymmetry index ≤ 6% indicated that the left and right structures were to be considered symmetrical, whereas a result > 6% was indicative of asymmetry. No asymmetry was found between the right and left sides. The presence of an increased condylar asymmetry index in a developing patient is a sign of altered skeletal growth and should be considered in the diagnostic process and treatment plan.

The patient is currently undergoing orthodontic treatment. The examination showed Angle ½ of II class malocclusion with maxillary constriction and lack of space for the central incisor. The overjet is 2 mm and the overbite is 3 mm. As part of the orthodontic treatment plan, removable appliance therapy was recommended. In addition, oral cavity sanitation was advised, along with appropriate oral hygiene instructions. Orthodontic treatment is intended to recreate space in the upper arch to replace the missing incisor. For this purpose, maxillary expansion appliances are used. The presence of both central incisors in the upper arch is important for the aesthetics of the smile and the function of chewing. An additional radiological examination is planned in order to assess the morphology of the palatal suture and the latero-lateral teleradiography for cephalometric analysis, in consideration of planning further treatment and restoring space for the incisor and implant-prosthetic treatment.

## 3. Discussion

Solitary Median Maxillary Central Incisor Syndrome (SMMCI) is an anomaly affecting structures located along the midline of the body. Numerous abnormalities of organs are primarily observed in the craniofacial region. The hallmark feature of the syndrome is the presence of a single maxillary central incisor, which develops in both the primary and permanent dentition and is limited exclusively to the maxilla [10]. It is a non-isolated congenital defect, which necessitates an interdisciplinary treatment approach and individualized care for each patient. According to scientific reports, phenotypic features characteristic of SMMCI can already be recognized in the neonatal period [11]. A child’s face may show distinct deviations from generally accepted norms, which can be crucial for early diagnosis of the condition. The treatment of a patient with SMMCI should involve a multidisciplinary team consisting of a dentist, orthodontist, speech therapist, and ENT specialist [12]. When diagnosing SMMCI, it is essential to exclude the following conditions: loss of one central incisor due to trauma or extraction, hypodontia (congenital absence of a tooth germ) on one side without, a midline-positioned incisor, supernumerary tooth (mesiodens), isolated dental anomalies without signs of midfacial dysmorphology, fusion of a primary or permanent incisor with a supernumerary tooth [4,13].

SMMCI is a rare congenital anomaly affecting the midline structures of the body. Its development is associated with various etiological factors, including genetic mutations, exposure to environmental influences, or other developmental disturbances. The exact pathomechanism of the disorder remains not fully understood [14]. The most likely cause of SMMCI is an abnormal migration of the mesoderm in the region of the frontonasal process and its failure to fuse along the midline between the 35th and 38th days of fetal development [15]. According to Lygidakis and Zatoński [6,13], the most important causes of SMMCI include:Missense Mutations in the Shh Gene, Located at Locus 7q36.Genetic factors related to maxillary incisor development. The presence of a solitary incisor may result from a developmental defect in tooth morphogenesis. Aberrations in genes responsible for the formation of maxillary incisors can lead to SMMCI.Gene expression during embryonic development. Disruptions in the timing and localization of gene expression can contribute to abnormal craniofacial skeletal formation, including SMMCI. According to case reports in the literature, spontaneous mutations can also occur.Mutations in Homeobox (HOX) genes, which regulate cell growth and differentiation during embryogenesis, including dental development.Mutations in genes controlling midline facial movements. Dysfunction of these genes during embryonic development can result in defects in the structure of the anterior facial region. SMMCI may be inherited in an autosomal dominant manner. It is also associated with Pallister-Hall syndrome, caused by a mutation in the GLI3 gene.

Although SMMCI may appear as an isolated feature, its presence can also be part of a broader syndrome, indicating the possibility of coexisting anomalies, particularly those within the holoprosencephaly (HPE) spectrum [16,17]. The inheritance pattern of HPE, like that of SMMCI, is autosomal dominant. HPE is a developmental disorder in which the brain fails to divide properly into two hemispheres, often associated with neurological impairment and craniofacial dysmorphism [18,19,20].

The most severe forms of holoprosencephaly (HPE) are lethal and often result in spontaneous miscarriage, whereas milder forms may present as SMMCI. Although individuals with SMMCI usually exhibit mild symptoms, they are considered to be within the HPE spectrum and may carry a risk of having children with more severe forms of HPE [21]. SMMCI has also been sporadically described in other syndromes not related to HPE, such as CHARGE syndrome, VACTERL association, ectodermal dysplasias, DiGeorge syndrome [22,23,24]. In the case of the described patient, the clinical features support the diagnosis of SMMCI.

The incisor in SMMCI differs from a normal central incisor in that the crown is symmetrical in shape. The primary clinical diagnostic criterion for SMMCI is the presence of a single central incisor located in the midline of the maxilla. In the examined patient, this was the key feature that led to the diagnosis of SMMCI. No significant craniofacial abnormalities described in the literature were observed in the patient’s facial structure. Craniofacial clinical features observed in patients with SMMCI include facial asymmetries, most commonly involving cleft palate, abnormal maxillary structure (with a V-shaped palatal arch), mandibular hypoplasia and/or malformations of the zygomatic bones [25,26], and lip deformities, typically affecting the upper lip. A flat, arched upper vermilion border without a clearly defined Cupid’s bow is often observed. Intraorally, this may present as absence of the upper labial frenulum and incisive papilla [27]. Nasal abnormalities are also encountered, particularly those involving the broad nasal bridge and/or rounded nasal tip.

In approximately 90% of cases, pyriform aperture stenosis and choanal atresia may also occur [1,2,28] and tooth abnormalities, including hypodontia and irregular tooth positioning in the oral cavity [29]. In the described patient, among the listed anomalies, palatal structural changes were observed. Based on the literature review by Nanni et al., McNamara, and Chandrasekaran D, in SMMCI, anomalies may also manifest in other organ systems: [16,30,31] cardiovascular system—congenital heart defects (in 25% of cases), including Tetralogy of Fallot (15%); urinary system—unilateral kidney agenesis; nervous system—chronic anterior pituitary insufficiency, brain malformations, and mild to severe intellectual disability (50%); auditory system—hearing impairments; visual system—strabismus, microphthalmia, and retinal abnormalities; respiratory system—various structural and functional abnormalities; skeletal system—scoliosis (14% of cases) and spinal and finger deformities; esophageal atresia—occurs in about 10% of cases; clavicle hypoplasia; olfactory dysfunction—anosmia (loss of smell); hormonal disorders—hypothyroidism; short stature—reported in 50% of patients.

However, only ~33% of cases are associated with confirmed growth hormone (GH) deficiency. In the remaining patients, short stature may be caused by other developmental or genetic factors [32]. The most serious symptom, often life-threatening, is breathing difficulty caused by abnormal anatomy of the upper airway, particularly stenosis of the nasal passages and the pyriform aperture, or choanal atresia [33]. In the examined patient, disordered breathing is present and is caused by very narrow nasal passages. Solitary Median Maxillary Central Incisor Syndrome may also be associated with various cognitive function impairments [34].

However, the described patient is developing normally in this regard. It is generally considered desirable to restore the esthetic appearance of the dental arch to its normal form with two normally shaped and positioned central incisors. Considering the patient’s age, the presence of mixed dentition, and maxillary constriction, treatment with a removable appliance was introduced, aiming to improve the inclination of the central incisor. Such a treatment plan is recommended in the literature by Barcelos et al. [35]. Other recommendations for treating patients with SMMCI syndrome can be found in the literature. Hall recommends starting therapy only during the permanent dentition period by expanding the palate transversely and distally moving the incisor to make space for prosthetic restoration of the absent incisor. Machado et al. recommend jaw expansion as the initial stage of treatment, followed by the use of a thin-arch appliance [36]. However, some authors recommend maxillary osteotomy prior to orthodontic treatment due to the lack of a palatal suture in the anterior palate [37].

### Summary

The diagnosis of Solitary Median Maxillary Central Incisor Syndrome (SMMCI) is often made around the age of 6, when the permanent central incisor begins to erupt. However, the presence of a single central incisor located in the midline in the primary dentition should raise concern among pediatricians and dentists, and a child showing such a sign should be referred for further evaluation and placed under the care of a pediatric dentist. Early diagnosis and recognition are crucial for assessing growth and development, as it may be a symptom of other severe congenital abnormalities. Monitoring facial and occlusal development is essential for achieving both aesthetic and functional rehabilitation. Because this anomaly may be associated with other developmental disorders, the child requires care from a multidisciplinary medical team.

## Figures and Tables

**Figure 1 children-13-00011-f001:**
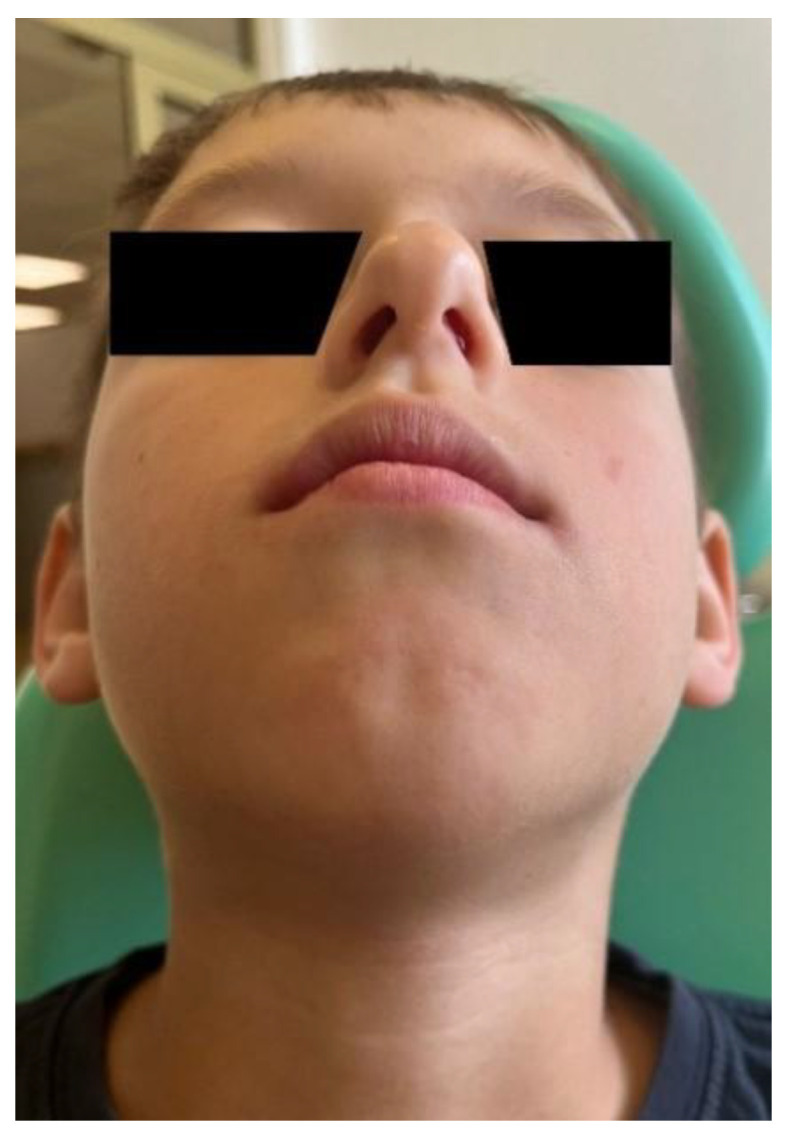
Clinical image showing the patient’s nostrils.

**Figure 2 children-13-00011-f002:**
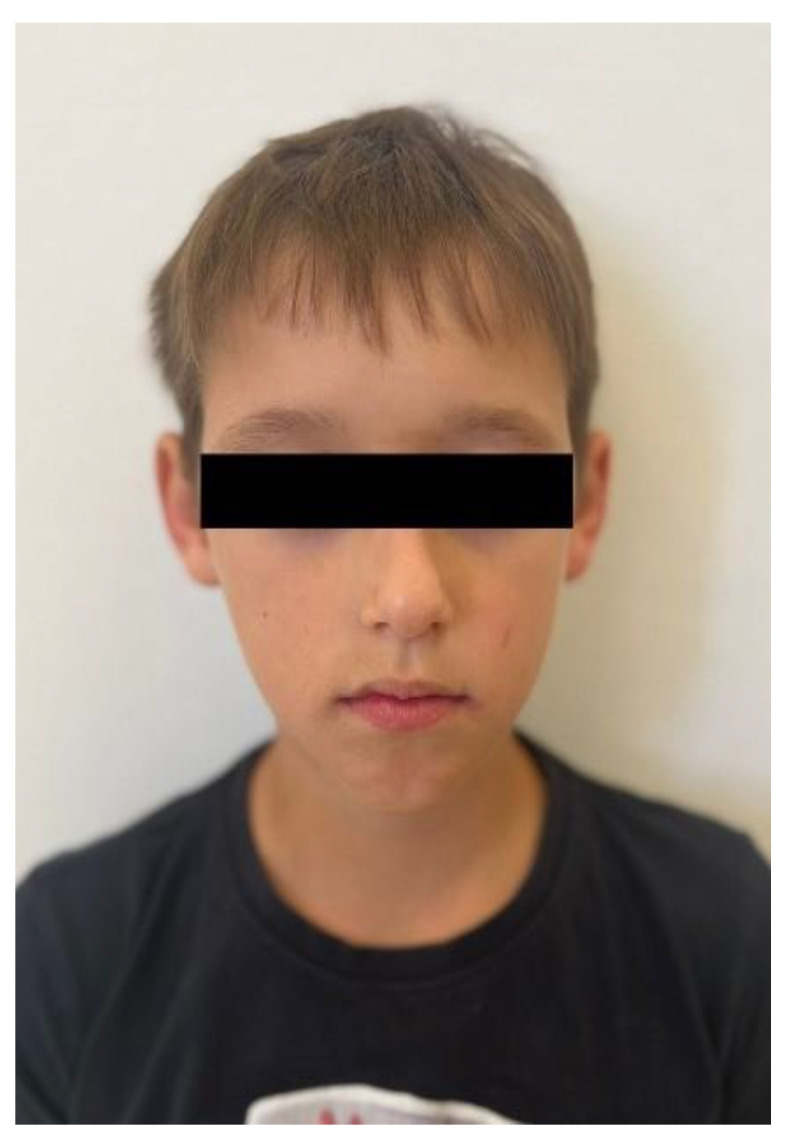
Frontal photograph of the patient’s face.

**Figure 3 children-13-00011-f003:**
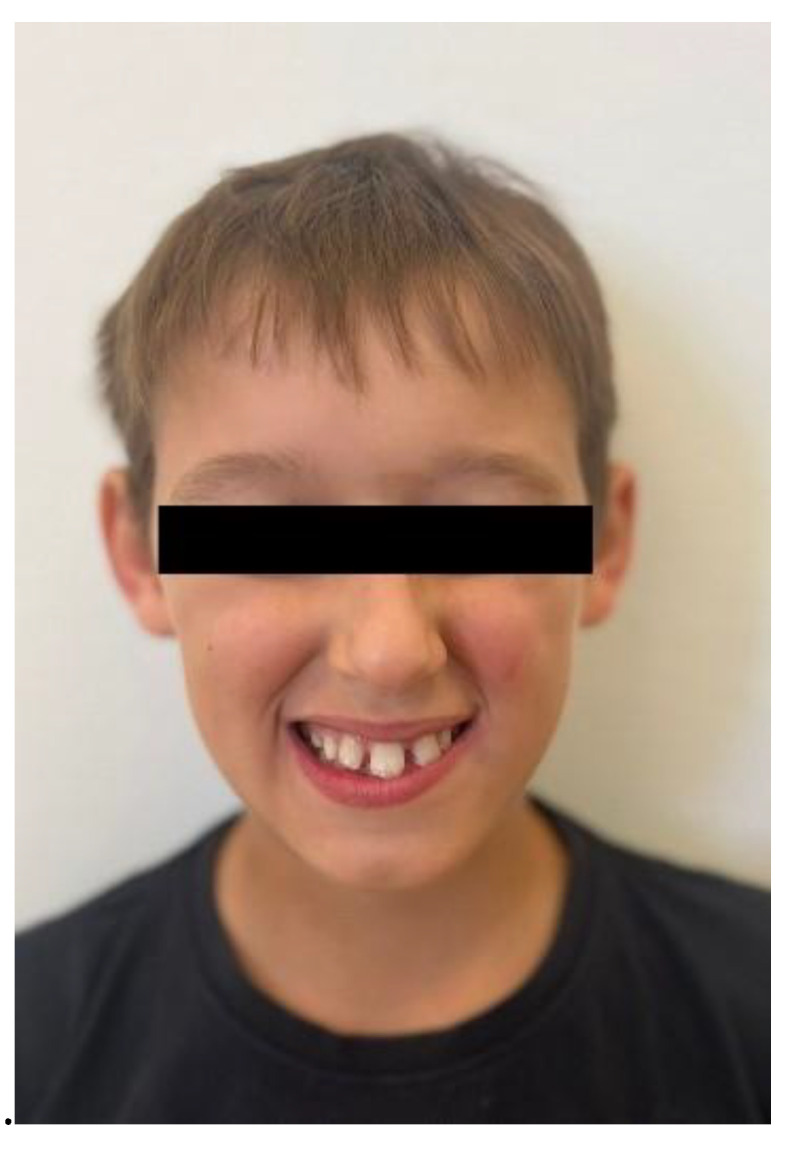
Frontal photograph of the patient’s face.

**Figure 4 children-13-00011-f004:**
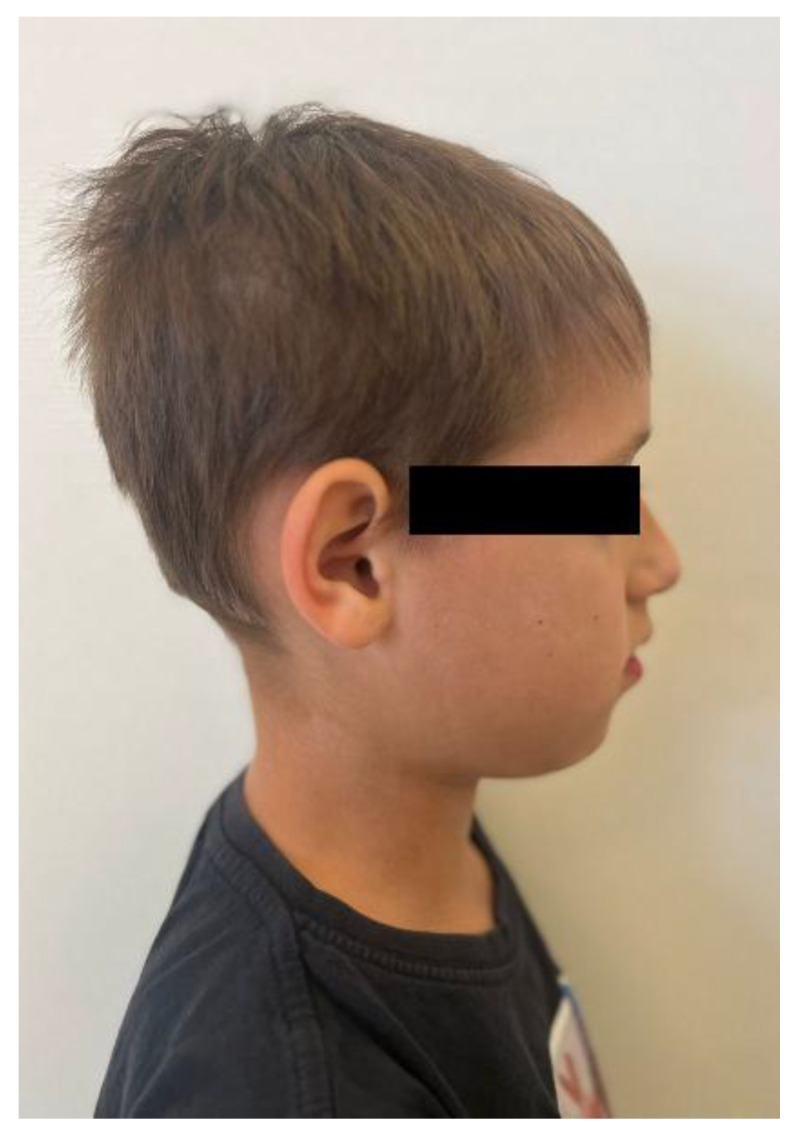
Right profile photograph of the patient.

**Figure 5 children-13-00011-f005:**
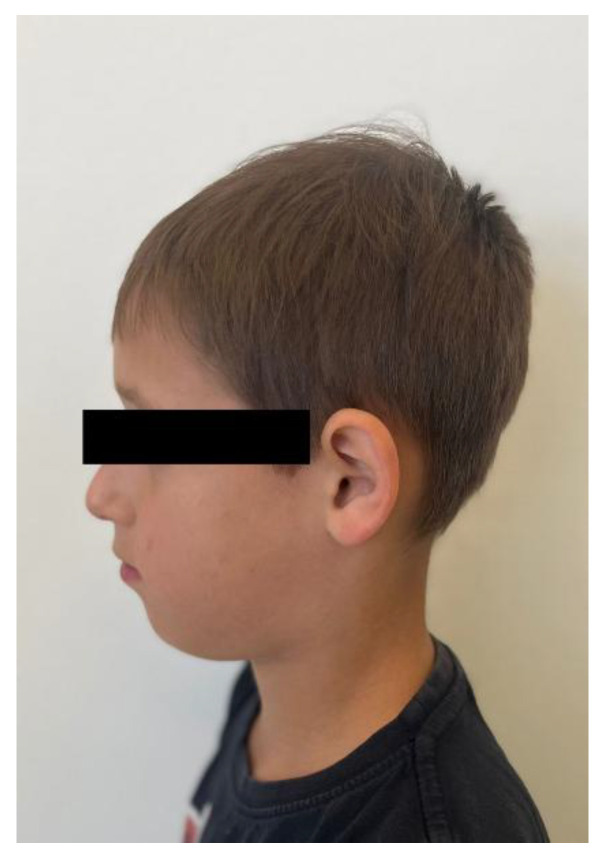
Left profile photograph of the patient.

**Figure 6 children-13-00011-f006:**
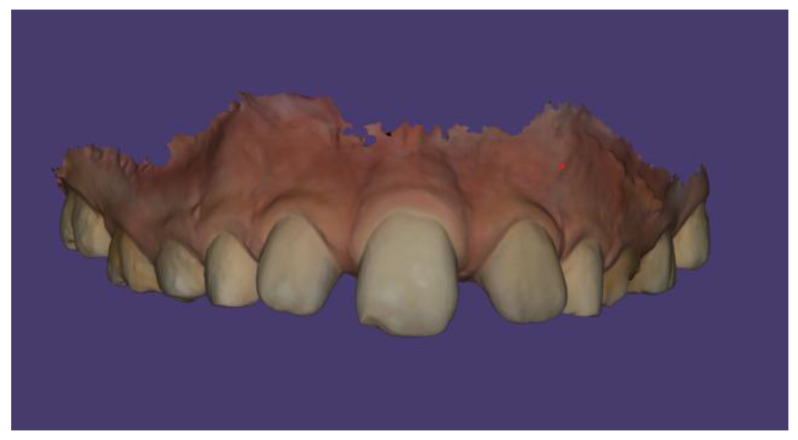
Intraoral scan of the maxillary arch.

**Figure 7 children-13-00011-f007:**
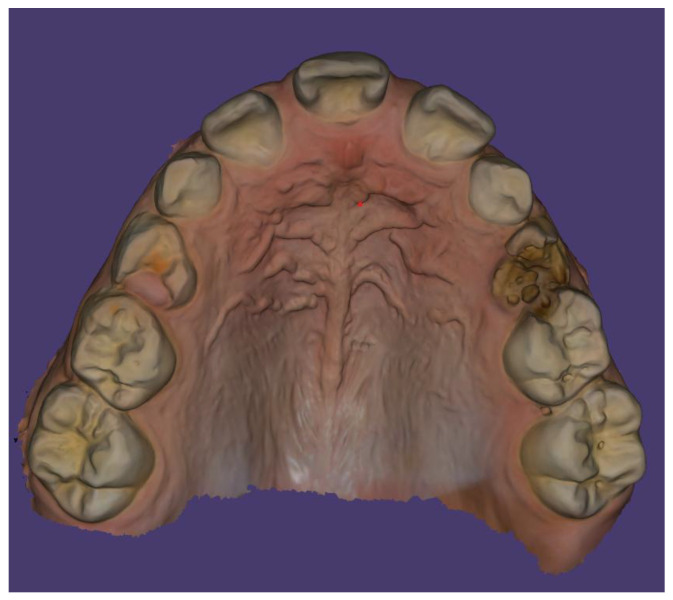
Intraoral scan of the maxillary arch—palatal surface.

**Figure 8 children-13-00011-f008:**
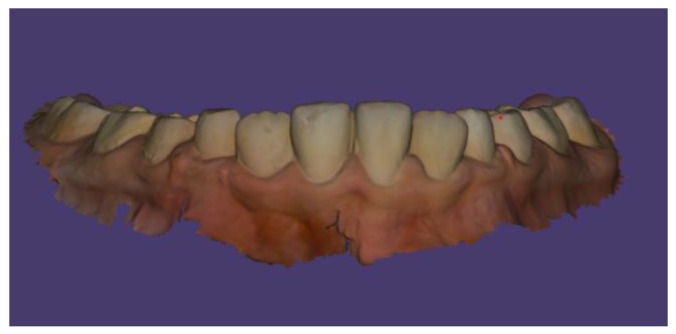
Intraoral scan of the mandibular arch.

**Figure 9 children-13-00011-f009:**
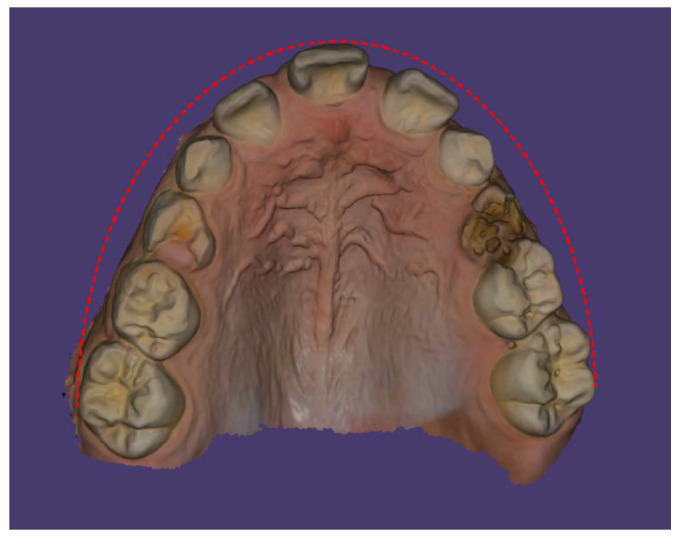
Shape of the maxillary dental arch.

**Figure 10 children-13-00011-f010:**
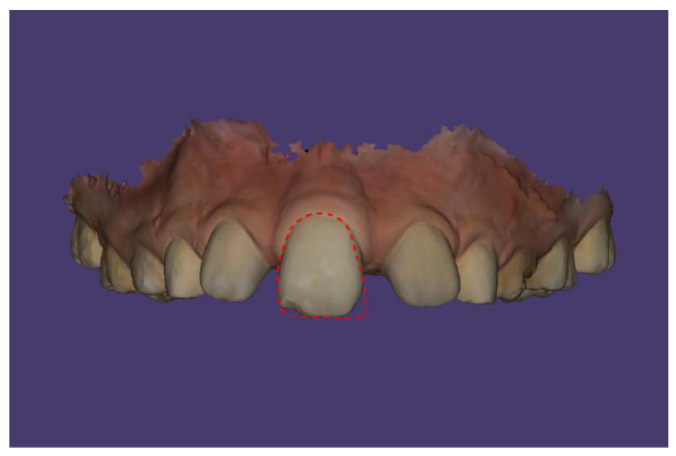
Comparison of the shape of the solitary central incisor with the outline of a right central incisor.

**Figure 11 children-13-00011-f011:**
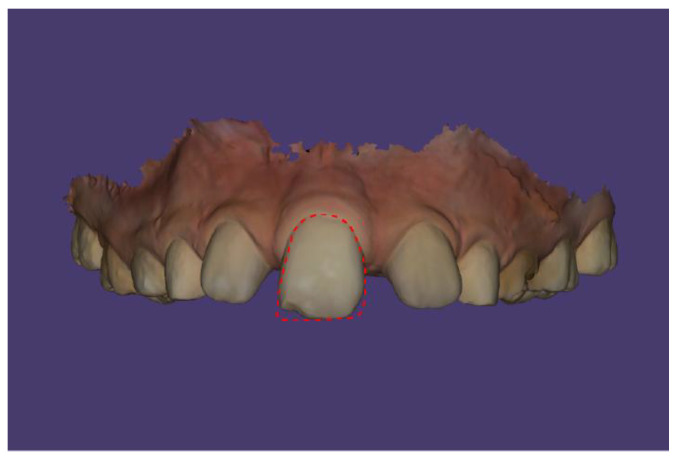
Comparison of the shape of the solitary central incisor with the outline of a left central incisor.

**Figure 12 children-13-00011-f012:**
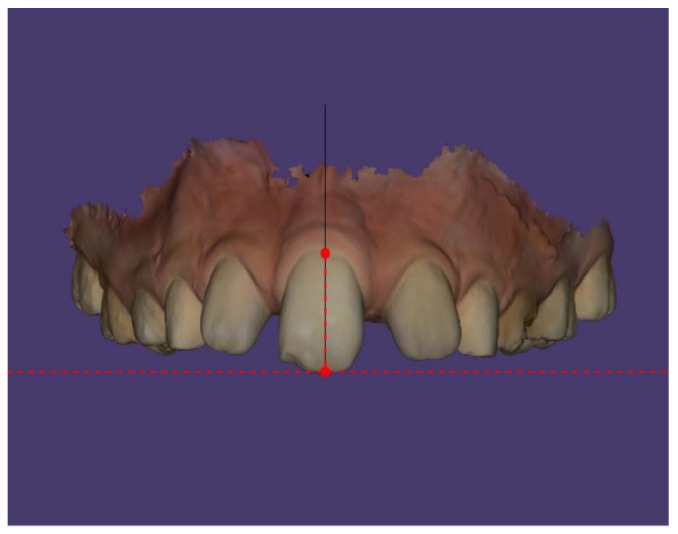
Tooth positioning according to Andrews’ key to occlusion.

**Figure 13 children-13-00011-f013:**
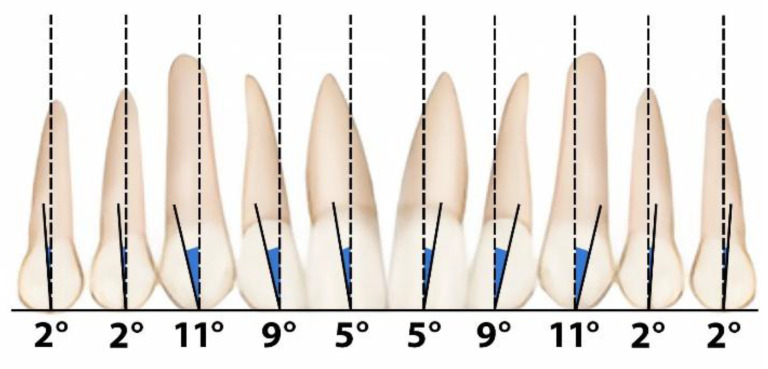
Andrews’ keys to occlusion.

**Figure 14 children-13-00011-f014:**
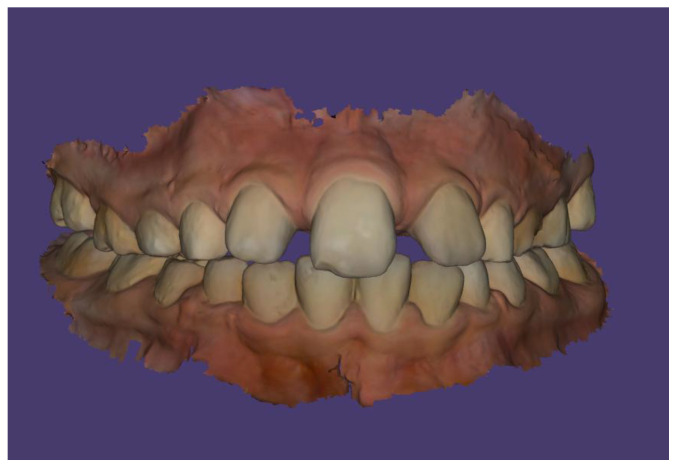
Scan of patient’s dental occlusion.

**Figure 15 children-13-00011-f015:**
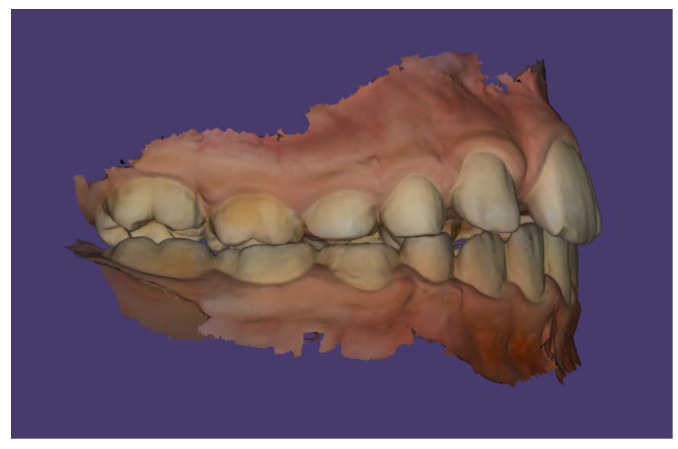
Scan of patient’s dental occlusion-right side.

**Figure 16 children-13-00011-f016:**
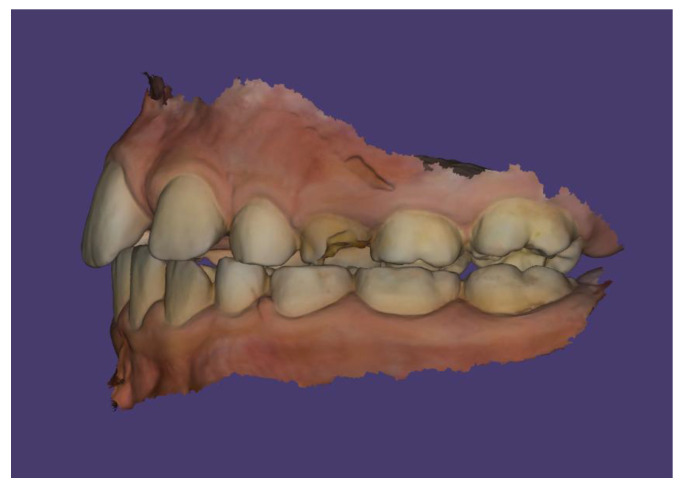
Scan of patient’s dental occlusion-left side.

**Figure 17 children-13-00011-f017:**
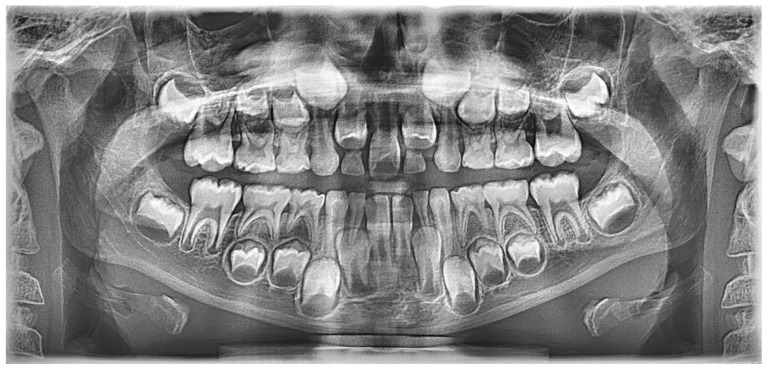
Panoramic radiograph of the patient.

**Figure 18 children-13-00011-f018:**
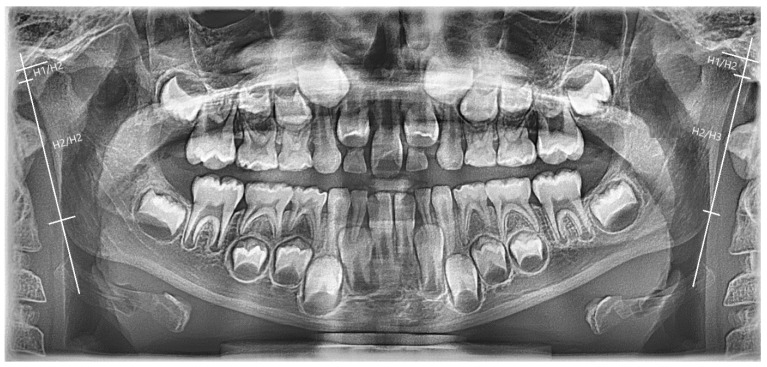
Panoramic radiograph of the patient. The figure illustrates the measurements of mandibular condyles and mandibular rami, H1/H2: height of the condyle; H2/H3: height of the ramus. Condylar asymmetry index 5%, Ramus asymmetry index 1.3%.

## Data Availability

The original data presented in the study are included in the article. Further inquiries can be directed to the corresponding author.

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
