# Peer review of "Children2026, 13(1), 11;https://doi.org/10.3390/children13010011"

_children, 2025, doi:10.3390/children13010011_

Round 1

Reviewer 1 Report

Comments and Suggestions for Authors

The manuscript titled "DIGITAL TECHNOLOGIES IN DIAGNOSING SOLITARY MEDIAN MAXILLARY CENTRAL INCISOR SYNDROME" is interesting. Here are some suggestions to improve it.

  • Interestingly the image of the palatal vault evidences an asymmetry between the two hemipalate being the right side narrower with respect to the left one: this should be described and the calculation of the condylar asymmetry added (doi: 10.1093/ejo/cjaf029).
  • even though the case report regards a child in mixed dentition it could be worth to describe the therapeutic proposal in the literature: this is interesting to the readers.
  • the risk of transmission to offspring with more severe forms of holoprosencephaly should be better highlighted to enable orthodontists, pediatric dentists and dentists to be aware of their sentinel role to prevent severe pathologies.

Author Response

I extend my gratitude for the insightful comments and suggestions, which have contributed to the improvement of the manuscript. I considered each point and revised the manuscript accordingly.

We agree that the palatal asymmetry is noteworthy finding that shoud be thoroughly described. As recommended we incorporated an analysis of condylar asymmetry based on advised method.

We also added therapeutic proposals from the literaturę as your suggestion was. Orthodontic treatment is intended to recreate space in the upper arch to replace the missing incisor. For this purpose, maxillary expansion appliances are used. The presence of both central incisors in the upper arch is important for the aesthetics of the smile and the function of chewing.

I admit that it is extremely important to emphasize the risk of genetic transmission to offspring  with holoprosencephaly. The genetic background places dental proffessionals, particularly pediatric dentists and orthodontists in crucial role.

Best regards Katarzyna Cieślińska

Reviewer 2 Report

Comments and Suggestions for Authors

 No cephalometric analysis

The Introduction states cephalometry is important for diagnosis, but no cephalometric tracing or measurements are included in this case. (Page 1)

Missing functional assessment

The child had:

  • mouth breathing
  • snoring
  • impaired suck reflex
    but no sleep study, no ENT functional scoring, and no airway metrics.

 “Materials and Methods” section is inappropriate for a case report

Case reports typically do not require a “Materials & Methods” section, and the current section reads more like a clinical workflow than methodology. (Page 2)

 No timeline of symptoms

Case reports must follow CARE guidelines, including an explicit timeline. Missing.

 No occlusal analysis details

Andrews’ keys are referenced (Page 6), but:

  • no full occlusion description
  • no overjet/overbite values
  • no molar/canine classifications
    Only Class II/I is mentioned.

 No scale bars or measurement indicators

Figures 7–9 compare tooth shapes, but neither calibration nor measurement annotations are visible.

Discussion too broad

The Discussion includes:

  • exhaustive literature review
  • syndromes (CHARGE, VACTERL, HPE, etc.)
  • systemic manifestations
    but minimal linkage to this patient’s specific findings.

 Overcitation

The Discussion references more than 50 studies for a single case.
MDPI Children prefers concise and relevant referencing.

Grammatical inconsistencies

Examples:

  • “He had to be bottle-fed…” (Page 2) — fragmented narration
  • “The boy exhibits reduced tone…” inconsistent tense use (Page 3)

Some sentences are too long and complex

Especially in the Discussion (Pages 8–9).

 Repetitive phrasing

E.g., “rare congenital anomaly affecting midline structures” appears 5+ times.

Author Response

Thank you for your insightful  feedback. We considered your comments and prepared revised version of the case report.

We agree that cephalometric analysis is a valuable diagnostic tool. However, due to the patient’s young age and the need to minimize radiation exposure, a cephalometric radiograph was not performed  at the time of the initial diagnosis. As the malocclusion is not a severe anterior-posterior defect, we decided not to take this image.

The symptoms of moth breathing and snoring were reported by parents and observed clinically. The patient remains under the care of an ENT specialist. The doctor has been informed of our diagnosis regarding the patient.

Thank you for your suggestion regarding the „Materials and Methods” section. When preparing our thesis, many articles describing case reports included this section, and we followed their example.

We aknowledge the importance of a timeline. A chronological account of patient’s care including symptoms, diagnosis and treatment plan is included in the article.

We expanded the description of the patient’s occlusion like overjet and overbite measurements. I hope this provides a more comprehensive occlusal analysis.

Figures 7-9 are intended to present only a comparison of the shape of the right and left incisors. Measurements of the central incisor are included in the text.

We revised the discussion. The numer of citation is 37 and it is difficult to exclude any of them, as each seems to be valuable for this description.

Thank you for bringing our attention to the grammatical inconsistencies. The manuscript was edited for grammar and style again. We reviewed it for repetitive phrasing and reworded sentences.

We believe that these revisions have improved the manuscript. Thank you again for your time.

Best Regards Katarzyna Cieślińska

Round 2

Reviewer 1 Report

Comments and Suggestions for Authors

The manuscript titled "DIGITAL TECHNOLOGIES IN DIAGNOSING SOLITARY

MEDIAN MAXILLARY CENTRAL INCISOR SYNDROME" has been improved, thank you.

One more effort to improve the data:

What is “Angle 1⁄2 of II class malocclusion”? The images show a neutral molar class due to the mixed dentition stage of the patient, but to properly plan the orthodontic therapy the latero-lateral teleradiography and cephalometry are essential diagnostic data. The risk of mistake due to the lack of cephalometry is worse than the radiological risk. For this reason, try to ask the patient to undergo a lateral teleradiography and add the cephalometry with the treatment plan.

If this is not possible, add, at least, the patients’ profile and the image of the appliance trying to explain the orthodontic plan.

Author Response

Thank you very much for your comments on the manuscript.
1/2 Angle's class  refers to the relationship between the buccal tuber of the first permanent molar in the maxilla and the buccal tuber of the first permanent molar in the mandible. At this stage of development (in mixed dentition), it is a normative occlusion. 

At this point, it wasn't possible to do a teleradiogram because of a lack of consent. We're planning to do this radiogram and cephalometric analysis later on in the treatment. My plan is to publish the exact course of treatment along with the restoration of the missing tooth.
Photos of the patient's profile have been added to the text.
Thanks a lot and  best regards,

Katarzyna Cieślińska

Reviewer 2 Report

Comments and Suggestions for Authors

It has been sufficiently improved to warrant publication in this Journal.

Author Response

Thank you very much for your positive review.

Best Regards, Katarzyna Cieślińska